# Frequent Acquisition of Glycoside Hydrolase Family 32 (GH32) Genes from Bacteria via Horizontal Gene Transfer Drives Adaptation of Invertebrates to Diverse Sources of Food and Living Habitats

**DOI:** 10.3390/ijms25158296

**Published:** 2024-07-30

**Authors:** Xiaoyan Cheng, Xuming Liu, Katherine W. Jordan, Jingcheng Yu, Robert J. Whitworth, Yoonseong Park, Ming-Shun Chen

**Affiliations:** 1Department of Entomology, 123 Waters Hall, Kansas State University, Manhattan, KS 66506, USA; xycheng325@live.cn (X.C.); xuming.liu@usda.gov (X.L.); jwhitwor@k-state.edu (R.J.W.); ypark@ksu.edu (Y.P.); 2Hard Winter Wheat Genetics Research Unit, Center for Grain and Animal Health Research, US Department of Agriculture, Agricultural Research Services, 4008 Throckmorton Hall, Kansas State University, Manhattan, KS 66506, USA; katherine.jordan@usda.gov; 3Department of Biochemistry and Molecular Biophysics, 141 Chalmers Hall, Kansas State University, Manhattan, KS 66506, USA; jingcy@ksu.edu

**Keywords:** GH32, glycoside hydrolase family 32, horizontal gene transfer, sugar metabolism

## Abstract

Glycoside hydrolases (GHs, also called glycosidases) catalyze the hydrolysis of glycosidic bonds in polysaccharides. Numerous GH genes have been identified from various organisms and are classified into 188 families, abbreviated GH1 to GH188. Enzymes in the GH32 family hydrolyze fructans, which are present in approximately 15% of flowering plants and are widespread across microorganisms. GH32 genes are rarely found in animals, as fructans are not a typical carbohydrate source utilized in animals. Here, we report the discovery of 242 GH32 genes identified in 84 animal species, ranging from nematodes to crabs. Genetic analyses of these genes indicated that the GH32 genes in various animals were derived from different bacteria via multiple, independent horizontal gene transfer events. The GH32 genes in animals appear functional based on the highly conserved catalytic blades and triads in the active center despite the overall low (35–60%) sequence similarities among the predicted proteins. The acquisition of GH32 genes by animals may have a profound impact on sugar metabolism for the recipient organisms. Our results together with previous reports suggest that the acquired GH32 enzymes may not only serve as digestive enzymes, but also may serve as effectors for manipulating host plants, and as metabolic enzymes in the non-digestive tissues of certain animals. Our results provide a foundation for future studies on the significance of horizontally transferred GH32 genes in animals. The information reported here enriches our knowledge of horizontal gene transfer, GH32 functions, and animal–plant interactions, which may result in practical applications. For example, developing crops via targeted engineering that inhibits GH32 enzymes could aid in the plant’s resistance to animal pests.

## 1. Introduction

Glycoside hydrolases (GHs, also called glycosidases) catalyze the hydrolysis of glycosidic bonds in polysaccharides [1]. The primary function of GHs is to release simple sugars from polysaccharides including glucans (polyglucose) and fructans (polyfructose). In addition, GHs are involved in other functions such as plant defense, signaling, and stress responses [2]. GHs are ubiquitous and functionally diverse. To date, more than 1,000,000 putative GH sequences have been identified and are classified into 188 families (https://www.cazypedia.org/index.php/Glycoside_Hydrolase_Families) (accessed on 1 May 2024). The common examples of GHs include cellulases, hemicellulases, amylases, pectinases, and fructosidases. One of the GH gene families is GH32, which contains glycoside hydrolases that hydrolyze the glycosidic bond between two or more fructose-containing carbohydrates (fructan), or between a fructose-containing carbohydrate and a non-carbohydrate moiety [3]. An example of a well-characterized GH32 enzyme includes invertases (also called β-fructofuranosidases) that hydrolyze sucrose in diverse species across multiple kingdoms ranging from the bacterium *Thermotoga maritima* [4], the yeast *Saccharomyces cerevisiae* [5], the plant *Arabidopsis thaliana* [6], and the insect *Bombyx mori* [7]. Other examples include exo-inulinases that hydrolyze β-(2 → 1) linked fructans (inulin) from the fungus *Aspergillus awamori* [8] and the plant *Cichorium intybus* [9]; and levanases that hydrolyze β-(2 → 6) linked fructans (levan) from the bacterium *Bacteroides thetaiotaomicron* [10]. The 3D structures of these enzymes suggest each of their interactions with ligands contains a catalytic triad necessary for enzymatic activity with consensus sequences “wmn**D**png”, “R**D**P”, and ‘**E**c” [7].

Despite fructans and their hydrolyzing enzymes, such as GH32s, being widespread across microorganisms, approximately only 15% of flowering plants contain significant amounts of fructans [11]. The most highly abundant plant species containing fructans are grown in dry environments or go through cold winters, suggesting fructan content might enhance their tolerance against drought and cold stresses. In fact, the evolution and diversification of plants containing fructans is speculated to have occurred in the mid-Tertiary era and again in the phases of the early Holocene epoch which are associated with dry climates [11]. Crops such as wheat, bananas, onion, garlic, artichoke, asparagus, and chicory contain a relatively high content of fructans in their roots, tubers, stems, tiller bases, and leaf sheath tissues [12]. Fructans are not used as an energy reserve for the majority of plant species and all the animal species. In fact, during evolution, animals lost GH32 genes and the ability to digest fructans using their own digestive systems. For those animals that live on plants containing significant amounts of fructans, an alternative way to use host fructans as a source of nutrients needs to be developed.

Horizontal gene transfer (HGT) is the phenomenon of acquiring genetic material by one species directly from a different species. It is often a means for the recipient to gain extra abilities, for example, digesting undigestible food [13]. HGT has been documented in a wide range of organisms from viruses to animals. Genes gained via HGT can provide certain advantages for the recipient organism. In some cases, a single HGT event can lead to drastic adaptation to otherwise hostile environments for the recipient. Some examples include the horizontal transfer of a lectin-like antifreeze gene, which allows several recipient fish species to live in an otherwise lethal environment [14]; the acquisition of antibiotic resistance genes allows a range of bacteria to gain drug resistance [15]; and the gaining of genes encoding plant cell wall–degrading enzymes enhances the ability of fungal parasites to attack hosts [16].

The most-documented horizontally transferred GH genes from prokaryotes to eukaryotes encode enzymes that digest cell walls, including pectinases [17] and cellulases [18], which enhances the ability of parasites to attack host plants. Recent evidence suggests that GH32 genes encoding levanases and inulinases are also potential HGT targets from prokaryotes to eukaryotes. The gaining of these enzymes could allow for the recipients to utilize otherwise undigestible nutrients. For example, an HGT-derived, sucrose-hydrolyzing GH32 gene allows the nematode *Globodera pallida* to utilize host-derived sucrose as nutrients [19]. Previously, our group discovered a single HGT event of a GH32 gene that led to the acquisition of a fructan metabolic pathway in a gall midge [20]. Here, we expanded our search for any horizontally transferred GH32 genes across all animal species with a sequenced genome deposited in Genbank. We detected a wide range of animal species that have independently acquired different GH32 genes from different sources. Substantial evidence suggests that the repeated acquisition of GH32 genes by various animal species may be a major driving force for species adaptation to diverse food sources and habitats.

## 2. Results

### 2.1. GH32 Genes in Diverse Animals

As of 31 December 2023, more than 3300 animal genomes are sequenced and deposited into GenBank. We searched these genomes and found that 940 (28%) of the animal species contain GH32-encoding genes. These species containing GH32-encoding genes are distributed into nine phyla, including Porifera (sponges, 1 species), Cnidaria (coelenterates, 3 species), Platyhelminths (flatworms, 1 species), Annelida (segmented worms 1 species), Mollusca (mollusks, 1 species), Tardigrada (water bears, 2 species), Rotifera (rotifers, 11 species), Nematoda (nematodes, 24 species), and Arthropoda (bugs, 895 species) (Table 1, Appendix A). The largest phylum containing GH32-encoding genes was Arthropoda, with 895 species distributed in 18 orders. Lepidoptera (moths and butterflies) is the largest order, with 730 species that contain GH32-encoding genes. Given each lepidopteran genome sequenced to date contains GH32 gene(s), the similarity of gene structures (all without any introns), and protein similarity > 80%, we selected 11 representative lepidopterans in this study. Table 1 lists the number of animal species per group containing GH32 genes, the number of GH32 genes found per species, and the total number of GH32 genes identified per group.

### 2.2. Conservation of the Catalytic Triads and 3D Structures

Despite an overall sequence identity of only 35–60% among the GH32 putative proteins identified from different animals, the catalytic triad sequences are highly conserved (Figure 1A, Appendix A). The consensus sequence at the active site of region 1 is XWZN**D**PNGZ, where Z represents a hydrophobic amino acid residue of either M, I, L, V, Y, or W; and X represents any amino acid residue. The consensus sequence of the active site of region 2 is XXZR**D**PXZZ, and the consensus sequence of active region 3 is XXZZ**E**CPXZ. Taken together, these active sites produced a catalytic triad of D, D, and E, which were conserved among 93.4%, 94.1%, and 99.2% of all the sequences at the three sites, respectively.

The three-dimensional structures of the identified proteins are also highly conserved. As shown in Figure 1B, five blades (marked as blades I, II, II, IV, and V) at the active site of the *Bombyx mori* β-fructofuranosidase were determined via the X-ray diffraction of the protein crystal [7]. All three-dimensional structures of the identified proteins predicted using AlphFold-2 (Figure 1B and Appendix A) contained these five blades, while some proteins contain some additional domains.

### 2.3. Structural Variation among Genes from Different Animals and Genes from the Same Animal

Large variation not only exists in the sequences of the putative proteins but also in the gene structure from different animal species and within some genes from the same species (Figure 2). First, the positions and distribution of introns in genes from different animals vary (Figure 2A). Second, the number of introns per gene also varies, ranging from zero (in 141 genes) to fourteen (in 7 genes). Third, the sizes of introns greatly differ, where most of the intron sizes are 100–200 nucleotides (Figure 2C), and some have intron sizes > 5000 bp.

Not only are the gene structures between animals different, but also they can be quite different within the same animal. For example, there are three GH32 genes in the genome of *Sinella curviseta*, and each of the genes has a different structure (Figure 2D). Two genes have two introns. However, the locations of the introns are different between the two genes based on the positions corresponding to their encoded amino acid sequences. The other gene has four introns. Genes with different numbers of introns from the same animal were also found in *M. destructor*, *Contarinia nasturtii*, *Bradysia coprophila*, *Bradysia odoriphaga*, *Cyphoderus albinus*, *Pogonognathellus flavescens*, *Yoshiicerus persimilis*, *Tomocerus vulgaris*, *Tomocerus qinae*, *Orchesella cincta*, *Folsomia candida*, *Thrips palmi*, and *Oedothorax gibbosus*.

### 2.4. Faster Diversification Once GH32 Genes Transferred from Bacteria to Animals

The proteins identified from various animal species along with best-matched bacterial sequences and selected plant and fungal proteins were subjected to a CLANS analysis. The results revealed that the GH32s from animal species are broadly divided into two major groups, except for some GH32s from nematodes and rotifers, which formed two additional small clusters (Figure 3). The two major groups are named GH32A, which includes the genes encoding proteins with levanase/inulinase activities, and GH32B, which includes genes encoding proteins with sucrase activities (Figure 3) [20]. The functions of the two minor GH32 clusters remain to be determined. Bacterial GH32s are widely distributed across all the groups. Interestingly, within each group, bacterial sequences are mainly clustered in the central region, whereas those from animals are located on the edge area of each cluster. This phenomenon suggests that GH32 genes have been diversifying more rapidly once transferred to animals than in their original bacterial donors.

### 2.5. Ultiple GH32-Gene Transfer Events to Animal Genomes from Diverse Bacteria

The GH32 predicted proteins are quite diversified, with sequence identity ranging from 35 to 60% from different species. The predicted protein sequences were blasted against the bacterial proteins deposited in Genbank. The best bacterial matches of these 242 proteins were proteins from 21 orders of bacteria (Figure 4A). The majority (approximately 80%) of the matches were from six orders including Bacillales (32%), Cytophagales (18%), Enterobacterales (10%), Chitinophagales (8%), Hyphomicrobiales (6%), and Lactobacillales (5%). The remaining ~20% of the matches were from the orders Halanaerobiales, Sphingobacteriales, Bacteroidales, Flavobacteriales, Streptomycetales, Burkholderiales, Gemmatales, Chloroflexales, and Phototrophicales, with each representing less than 3%.

The relationship between potential bacterial donors and animal recipients appears complex. GH32 proteins from a single animal group (order) with more than two species all had the best matches to proteins from at least two different bacterial orders (Figure 4B). For example, 35 GH32 proteins from seven species of dipterans had the best matches to proteins from seven orders of bacteria. Six GH32 proteins from two species of water bears had the best matches to proteins from five orders of bacteria. More surprisingly, different GH32 proteins even from a single animal species had the best matches to proteins from different orders of bacteria. Figure 4C shows that 12 (5%) animal species from eight (38%) groups had the best matches to proteins from different orders of bacteria. For example, the springtail, *Sinella curviseta*, has three GH32 genes and each of the three encoded proteins had the best matches to proteins from different orders of bacteria. The 12 predicted GH32 proteins from the dipteran, *Contarinia nasturtii*, had the best matches to proteins from five different orders of bacteria.

Multiple best bacterial matches of animal GH32 proteins suggest that the GH32 genes from these animals could have been derived independently from different bacterial donors. The GH32 predicted proteins from different types of animals clustered with GH32 proteins from different types of bacteria based on a phylogenetic analysis (Figure 5). For example, GH32 proteins from beetles are distributed in four different clusters together with proteins from different orders of bacteria. GH32 proteins from wasps are distributed in five clusters. GH32 proteins from other groups of animals including nematodes, dipterans (flies), spiders/mites, rotifers, and water bears are also distributed to different clusters. The only exceptions are the GH32 proteins from moths/butterflies, which are found in a single cluster with bacterial proteins. Indeed, GH32 genes from moths/butterflies share the same gene structure without any introns, indicating that they may share the same origin before diversifying into different species.

### 2.6. Impact of Derived GH32 Genes on Sugar Metabolism in the Recipient Animals

We hypothesized that the acquisition of GH32 genes by animals might provide these animals the ability to digest fructans. Initial metabolite profiling indicated changes in the abundance of both fructose and glucose in the tissues of animals containing GH32 genes (Figure 6A). For example, relatively high levels of fructose were detected in Hessian fly feeding larvae, likely derived from the digestion of ingested fructans via the GH 32 enzymes. In contrast, virtually no fructose was detected in the Hessian fly’s close relative the fungus gnat, *Bradysia* spp, which based on PCR results does not contain any GH32 genes. Interestingly, we detected low levels of fructose in the pea aphid, which also contains no GH32 genes [22]. It is possible that the low levels of fructose may have been derived from the high concentrations of sucrose in its phloem diet. We also detected low levels of fructose in *Manduca sexta* larvae, whose genome contains several GH32 genes. In comparison, the levels of glucose were within a range in all the animals except pea aphids, which contained very high levels of glucose (Figure 6A). We think this is likely due to the high levels of sucrose ingested from its phloem.

To further examine the potential impact of HGT-derived GH32 enzymes on sugar metabolism, we selected four representative insect species to compare the absolute concentrations of fructose, difructose, and glucose. Difructose is the last intermediate in the digestion of fructans, such as levan and inulin, before releasing fructose. We compared the levels in the Hessian fly; wheat stem sawfly, *Cephus cinctus*, (which feeds on the same tissue of wheat as the Hessian fly); the fungus gnat; and the red flour beetle, *Tribolium castaneum*, (which feeds on wheat grains or flour). According to our research, only the Hessian fly genome contains the presence of GH32 genes [23,24]. We detected a large amount of fructose in Hessian fly larvae, and essentially no fructose in the other three species (Figure 6B). Similarly, a high level of difructose is present in Hessian fly larvae, but very little to zero is present in larvae of the wheat stem sawfly, fungus gnat, or red flour beetle. In contrast, lower levels of glucose were observed in the Hessian fly and sawfly larvae compared to the levels in the fungus gnat and red flour beetle larvae.

## 3. Discussion

Horizontally transferred genes from a different species are common in animal species. The transfer of GH genes from bacteria to animals is one of the main forces driving animal evolution and adaptation. Previous studies have mainly focused on horizontally transferred GH genes encoding enzymes capable of hydrolyzing cell walls, such as pectinases, cellulases, and hemicellulases [16,17,25]. The acquisition of cell wall-hydrolyzing enzymes enhances the ability of animal species such as nematodes and other parasitic species to attack host plants more effectively [19]. Recently, it was reported that genes encoding GH32 enzymes were horizontally transferred from bacteria to animals [19,20,26]. This study confirmed many animal species have obtained GH32 genes via horizontal gene transfer. It is possible that the acquisition of GH32 genes by animal species could not only be to gain the ability to attack host plants but also to access nutrients from hosts. Fructans, such as levan and inulin, are nonstructural storage sugars used as alternative energy reserves to glucans in various microorganisms and in certain plant species due to their roles in both drought and cold tolerance [27,28]. In contrast, animals and most other plants only use glucans, mainly glycogens and starch, as energy reserves. Animals cannot access the nutrients of fructans due to the lack of GH32 enzymes. We speculate that the gaining of GH32 genes via horizontal gene transfer allows the recipient animals to use the otherwise inaccessible nutrients of storage fructans.

A striking feature of horizontally transferred GH32 genes from bacteria to animals in comparison with other horizontally transferred genes is the frequency and independence associated with multiple transfer events from diverse donors to diverse recipients. As shown in Figure 4, most of the GH32 genes do not have an orthologous relationship among the genes from related animal species. We highlighted GH32 proteins from different beetles that formed different clusters with proteins from different orders of bacteria. Similar phenomena exist with GH32 proteins from other animal groups, such as wasps, flies, and rotifers. These observations suggest multiple gene transfer events have occurred independently even among sister species. In some cases, different GH32 genes from the same animal species were derived via different transfer events from different donors. An example includes the three genes from *S. curviseta* (Figure 1C and Figure 3D), which were apparently obtained from three different donors. This phenomenon is quite different from the horizontal transfer of other GH gene families from bacteria to animals. For example, the transfer of a pectinase gene (a GH28 member) from a bacterium to an insect preceded the diversification of the suborder of Polyneoptera, and the transferred gene has been retained in many herbivorous species of this suborder [29]. A perfect orthologous relationship exists among the horizontally transferred cellulase genes (GH5 family members) identified from multiple cerambycid beetles including *Apriona japonica* and *A. glabripennis*, suggesting that a single transfer event happened preceding the diversification of the beetles into different species [25]. A gene (a member of the GH43 family) encoding an α-arabinofuranosidase transferred from a bacterium to the springtail, *Folsomia candida*, is also closely related to genes found in other springtails including *Orchesella cincta* and *Allacma fusca* [30].

The reason(s) for the frequent and dynamic transferring of GH32 genes from bacteria to animals remains unknown. One possible explanation is that there are different types of fructans that require different enzymes to digest them. For example, the fructan inulin has mostly or exclusively the (2 → 1) fructosyl–fructose linkages, whereas levan has mostly or exclusively the (2 → 6) fructosyl–fructose linkages, depending on different types of plants [31]. There are two different ways to gain enzymes that digest substrates with different linkages. One way is to duplicate the same gene and then diversify the duplicates to gain different substrate specificities. This has apparently happened in the gall midge, *Mayetiola destructor*, where a single horizontally transferred gene duplicated into more than 10 copies, resulting in functional proteins with enzymatic activities towards different substrates [20]. The accelerated diversification of GH32 genes once they are transferred to animals from bacteria (Figure 3) also supports the possibility of GH32 genes acquiring different enzymatic specificity via diversification in the recipient animals. Another way to gain the ability to digest different substrates is to acquire genes encoding enzymes with different substrate specificities directly from other species via multiple horizontal gene transfer events. This might have happened in most animal species with transferred GH32 genes since there is no orthologous relationship between the GH32 genes even among closely related animal species.

Another interesting phenomenon associated with GH32 genes in animals is that GH32 genes can be present in one animal species but absent in a closely related species. For example, multiple GH32 genes are present in the Dipterans, *M. destructor* and *Contarinia nasturtii*, but not in related species such as *Drosophila melanogaster*, *Bactrocera dorsalis,* or *Orseolia oryzae*. This could be due to approximately only 15% of plant species containing storage fructans and the plant species with fructans are not necessarily genetically related. For example, rice and wheat are genetically related cereals, wheat stems contain significant amounts of fructans [32] while rice stems contain little fructans [33,34]. Even in the same plant species, one tissue contains large amounts of fructans but other tissues may not. For example, wheat stems contain relatively high levels of fructans, but mature wheat grains contain very few fructans [35,36]. If two animal species share the same origin but were differentiated to live on two different but genetically related host plants, one of which has fructans and the other does not, the animal species feeding on the plant with fructans gained an evolutionary advantage by acquiring GH32 genes, but the other feeding on plants without fructans does not need to acquire or retain any GH32 genes.

The acquisition of GH32 genes in some recipient animals has an advantageous impact on their metabolism. For example, the combined abundance of fructose and difructose exceeds the abundance of glucose in Hessian fly larvae (Figure 6B). During the long course of evolution, metabolism in animals was optimized to use glucose and trehalose (di-glucose) as blood sugars and major fuel for energy and metabolic intermediate production. In mammals, fructose can only be metabolized in certain tissues [37]. In fact, too much fructose in mammals can trigger the release of an intracellular alarm signal, which results in the organism going into a so-called ‘safety mode’, leading to multiple potential health issues [38]. The mechanism of how GH32-recipient animals metabolize the increased fructose efficiently needs to be investigated. Even though glucose and fructose have the same chemical composition (C6H12O6), they cannot be converted to each other directly in a reversible chemical reaction in living organisms. Rather, fructose is converted to glucose through very costly, multiple chemical reactions via the aldolase pathway [39]. One hypothetical way to use elevated fructose efficiently without causing the disruption of sugar metabolism is to use the oligomers and polymers of fructose as storage sugars directly as in some plants without converting them into glucose and then glucose polymers such as glycogen. Initial evidence of GH32 genes expressed in non-digestive tissues supports such a hypothesis. In Hessian fly larvae, several GH32 genes are expressed at high levels in fatty bodies and Malpighian tubes, and the enzymatic activity of GH32 enzymes is also detected in these tissues [20]. The fructans levan and inulin are detected in non-digestive tissue, as well as non-feeding stage (eggs) [20]. These pieces of evidence suggest that animals may directly use fructans as storage sugars. Further studies are needed to reveal how fructans ingested from plants are transported to other non-digestive tissues and/or how fructans are synthesized in non-digestive tissues with fructose released from ingested fructans via the action of GH32 enzymes. The Hessian fly genome does not contain genes encoding enzymes for fructan synthesis. However, genes encoding enzymes for fructan synthesis were found in bacteria associated with Hessian flies. It is possible that symbiotic bacteria help to synthesize fructans from GH32 enzyme-released fructose in the non-digestive tissues and non-feeding stages of the Hessian fly. Further research is needed to explore these possibilities.

How GH32 genes were transferred from various bacteria to numerous animal species remains unknown. The GH32 genes are in bacterial genomes without any obvious transposon elements around them. Presumably, there were only two ways, either direct gene fragment transfer, or a gene integrated into a vector (plasmid or phages) first and then transferred into an animal germ cell via the vectors. Further research needs to be conducted to delineate the transfer mechanisms. Similarly, the genetic mechanisms driving the expression of *GH32* genes in different tissues of the recipient animal species remain unclear. It is known that some promoters derived from certain transposons or viruses can enable the transcription of horizontally transferred bacterial genes in eukaryotes [40,41]. However, the analyses of nucleotide sequences in the promotor regions of the GH32 genes in animals revealed no evidence of potential transposon- or virus-originated elements or sequences from bacteria, indicating that the cis-elements driving the expression of the horizontally transferred bacterial genes in different tissues of the recipient animals may have arisen by some de novo mechanism.

In conclusion, we identified a large number of GH32 family genes in a wide range of organisms living on different food sources and habitats. Phylogenetic analyses suggest that GH32 genes in different animals were obtained via multiple gene transfer events independently from diverse bacterial donors. The predicted three-dimensional structures and the catalytic triads of the proteins encoded by the identified GH32 genes are highly conserved despite the overall low sequence similarity among these proteins. The conservation of the three-dimensional structures and the presence of the catalytic triads in nearly all the identified proteins suggest these GH32 enzymes are functional. The presence of GH32 genes in animal genomes appears to have a profound impact on sugar metabolism in the recipient animal species.

## 4. Materials and Methods

### 4.1. Gene Identification

We used the amino acid sequence of the GH32 β-fructofuranosidase (Genbank accession number ABN04092) from the bacterium *Bifidobacterium longum* to search the nucleotide sequence databases in Genbank with tblastn [42] because its structure is known. We searched against the whole-genome shotgun contigs (WGS) database since it contains more sequenced genomes than the nonredundant (NR) database. The identified WGS GH32 genes were compared with the ones in the NR database if the sequences were available. Due to the large number of genome sequences available in the WGS database, we performed multiple searches limited to the genomes of one phylum, one class, or one order of animals to increase the computational blast efficiency. A blast hit was considered a match to GH32 if the e-value was ≤0.05, and sequence identity was ≥30% with gaps ≤10%.

Once a potential gene was identified from the blast analysis, the candidate gene sequence was extracted from Genbank. Intron–exon boundaries were determined by aligning DNA sequences with protein sequences following the role of the junction consensus of intron donors and acceptors [43]. If a DNA sequence contained a small deletion or mutation that could disrupt the translation of the DNA sequence into a full-length protein, primers were designed to amplify that region, and PCR products were sequenced to check if there were errors in the original sequence.

### 4.2. Identification of the Catalytic Triads

The locations of the catalytic triads in a protein were determined by aligning individual sequences with the GH32 enzyme (Genbank accession: NP_001119721) from *B. mori* [7]. The location of a triad region identified from an alignment was further confirmed by comparing its location in the corresponding catalytic site in its three-dimensional structure predicted using AlphaFold-2 [44]. The triads in each protein were marked with different colors in the protein sequence in Appendix A.

### 4.3. Prediction of Three-Dimensional Structures of the Identified Proteins

The three-dimensional structures of the identified proteins were predicted using AlphaFold version 2 [44]. The predicted structures are given in Appendix A.

### 4.4. Best Bacterial Protein Matches of the Identified Animal GH32 Proteins

Translated protein sequences from the genes identified from various animals were used to search against the bacterial proteins contained in Genbank using BlastP [42]. The first hit of the bacterial sub-database was taken as the best bacterial match and the bacterial protein was extracted from Genbank for downstream phylogenetic and other analyses.

### 4.5. CLANS and Phylogenetic Analyses

The 242 GH32 proteins predicted based on genes identified from various animals along with 133 nonredundant bacterial proteins, which were the best matches to the identified animal proteins, were used for the analyses of protein groups and diversification. In addition, three GH32s from wheat (*Triticum aestivum*), four GH32s from *Arabidopsis thaliana*, and five GH32s from various fungi, including *Kluyveromyces marxianus*, *Fusarium sarcochroum*, *Glonium stellatum*, *Daedalea quercina*, and *Brettanomyces bruxellensis*, were also included for the analysis. The analysis was carried out using CLANS, a java-based application for visualizing protein families [21], with default parameters. The taxonomic groups were labeled with different colors.

The same set of proteins was used for the phylogenetic analysis. The phylogenetic analysis was conducted based on the neighbor-joining approach [45] using Molecular Evolutionary Genetic Analysis version 11 (MEGA11) with 500 bootstrap tests [46]. All the protein sequences used in this analysis are listed in Appendix A.

### 4.6. Metabolic Profiling and Concentration Determination

Metabolic profiling was carried out to measure soluble metabolites via a commercial contract by the company Metabolom (Durham, NC, USA). Briefly, the samples were processed through three steps: extraction, derivatization, and analysis. Before extraction, recovery standards were added for quality control purposes. For extraction, 0.25 mL of precooled (–20 °C) solvent containing water, acetonitrile, and isopropanol (2:3:3, *v*/*v*/*v*) was added to a sample consisting of 5 mg of frozen ground insect tissues. The samples were agitated for 5 min at 4 °C on a chilling and heating dry bath. The samples were then centrifuged for 2 min at 13,200 rpm, and 0.5 mL of the supernatant was transferred to a clean tube. The samples were dried via a speed vacuum. For sample derivation, 20 μL of a methoxyl-amine hydrochloride in a pyridine mixture (40 mg/mL) was added to each sample. The mixture was mixed in a dry bath for 30 min at 40 °C, followed by the addition of 180 μL of N-methyl-trimethylsilylacetamide (MSTFA). The samples were then heated at 40 °C for 30 min in an orbital shaker. LC/MS analysis was conducted on a platform based on a Waters ACQUITY UPLC and a Thermo-Finnigan LTQ mass spectrometer, which consisted of an electrospray ionization (ESI) source and a linear ion-trap (LIT) mass analyzer. The animal extracts were split into two aliquots, dried, and then reconstituted in acidic or basic LC-compatible solvents, each of which contained 11 or more injection standards at fixed concentrations. One aliquot was analyzed using acidic positive ion-optimized conditions and the other using basic negative ion-optimized conditions in independent injections with separate columns. The extracts reconstituted in acidic conditions were gradient eluted using water and methanol, both containing 0.1% formic acid, whereas basic extracts were eluted using water/methanol containing 6.5 mM ammonium bicarbonate. The MS analyses alternated between MS and data-dependent MS2 scans using dynamic exclusion.

The compounds were identified by comparison to library entries of purified standards or recurrent unknown entities. The identification of known chemical entities was based on a comparison to the metabolomic library entries of purified standards, which consisted of over 1000 commercially available purified standard compounds. The combination of chromatographic properties and mass spectra gave an indication of a match to the specific compound or an isobaric entity. A variety of curation procedures were carried out to ensure that a high-quality data set was made available for statistical analysis and data interpretation.

Sugar extraction and quantification were performed on an in-house system with samples prepared following the procedure described previously [47]. Briefly, for the quantification of fructose, difructose, and glucose, ~20 mg fresh animal samples were collected into 1.7 mL microcentrifuge tubes separately and immediately frozen in liquid nitrogen. The frozen samples were ground using a mortar and pestle in liquid nitrogen. Tissue powder was suspended into 200 µL of 75% ethanol, and then incubated for one hour at 4 °C. Following incubation, the samples were centrifuged at 12,000× *g* for 15 min at 4 °C. The supernatants were transferred to new 1.7 mL microcentrifuge tubes and then dried in a Refrigerator CentriVap Vacuum Concentrator (Labconoco, Kansas City, MO, USA). The dried extracts were dissolved into 200 µL of LC-MS grade water and then filtered with a 0.2 µm Captiva PES Filter Vial (Agilent, Santa Clara, CA, USA) to remove high-molecular-weight substances. The resulting extracts were separated and quantified by passing through an Agilent InfinityLab Poroshell 120 HILIC-Z column (2.1 mm × 150 mm, 2.7 µm on an Agilent 1290 Infinity II Liquid Chromatography (LC) system combined with an Agilent G6545A Q-TOF mass spectrometer (LC-Q-TOF) (Agilent, Santa Clara, CA, USA). The concentrations of specific sugars were determined by comparing them with the corresponding individual standards. The LC-Q-TOF conditions and parameters were as described by Mack and Wei [48] and Dai and Hsiao [49].

Data were analyzed using the SAS software. Multiple comparisons of the means of different treatments were computed using Fisher’s least significant difference (LSD) (SAS Institute, Inc., Cary, NC, USA). The data sets with LSD significance were reanalyzed through a Dunnett test to avoid a type I error. Additionally, *t*-tests were conducted to determine whether the unknown means for the two samples were different.

## Figures and Tables

**Figure 1 ijms-25-08296-f001:**
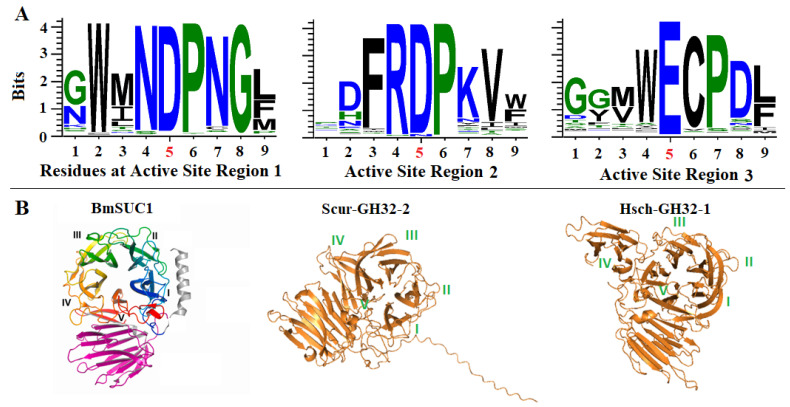
Conservation in the three triad sequence regions and the overall three-dimensional structures of the predicted proteins. (**A**). Logos show consensus sequences at the three active site regions. The residues D, D, and E at position 5 in each graph are the three triads. (**B**). The three-dimensional structures of three representative proteins. The structure of the protein BmSUC1 was determined by X-ray diffraction [7]. The structures of the proteins Scur-GH32-2 and Hsch-GH32-1 were predicted using AlphaFold-2. The five critical blades for catalytic activity are marked with I, II, III, IV, and V in each structure. The three-dimensional structures of other identified proteins are given in Appendix A.

**Figure 2 ijms-25-08296-f002:**
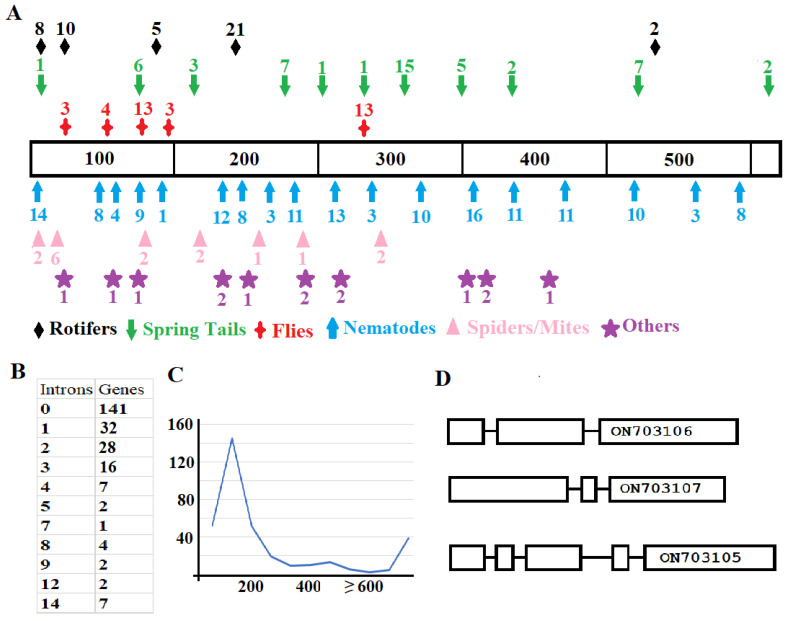
Variation in the structures of the identified genes. (**A**): The positions of the introns in genes from different types of animals. The middle bar indicates the positions of amino acids where an intron/exon boundary is located relative to the protein. The longest protein contains 538 amino acid residues. The types of animals are marked with different colors and symbols. The number of genes with an intron at the same position is given either above or underneath each symbol. (**B**): Genes with different numbers of introns. (**C**): The size distribution of introns in different genes. The *X*-axis indicates the base pairs of an intron whereas the *Y*-axis indicates the number of genes. (**D**): Different numbers and position of introns in the three genes from the springtail, *Sinella curviseta.* Genbank accession numbers are given in the last exon of each gene.

**Figure 3 ijms-25-08296-f003:**
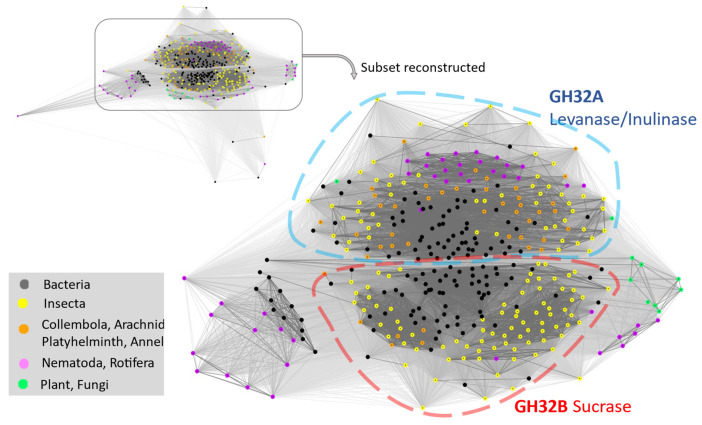
A two-dimensional map for GH32 groups of sequences from various taxa. A total of 242 eukaryotic proteins and 133 bacterial proteins were included. Default parameters in the software CLANS [21] were used and different taxonomic units are depicted by the colors in the figure legend. A subset of the upper left map was reconstructed in the map on the lower right. The dotted lines divide the two major clusters of GH32A levanase/inulinase and GH32B sucrase groups. Two minor groups containing GH32 proteins from nematodes and rotifers are located on the right and left sides of the two major groups.

**Figure 4 ijms-25-08296-f004:**
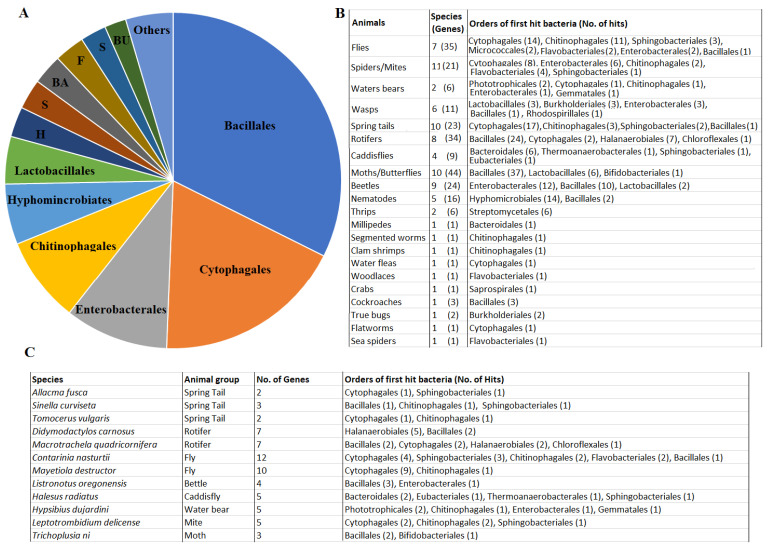
Predicted GH32 proteins from different animals had the best matches to proteins from different orders of bacteria in Genbank. (**A**): The distribution (proportion) of bacterial orders with best matches to the identified GH32 proteins from animals. H, S (brown), BA, F, S (blue), and BU represent bacterial orders Halanaerobiales, Sphingobacteriales, Bacteroidales, Flavobacteriales, Streptomycetales, and Burkholderiales. Other bacterial orders (Others) included Bifidobacteriales, Micrococcales, Saprospirales, Eubacteriales, Thermoanaerobacterales, Rhodospirillales, Gemmatales, Chloroflexales, and Phototrophicales. (**B**): GH32 proteins from the same type (order) of animals had the best matches to proteins from multiple orders of bacteria. (**C**): Different GH32 proteins from a single species of animals had the best matches to proteins from different orders of bacteria.

**Figure 5 ijms-25-08296-f005:**
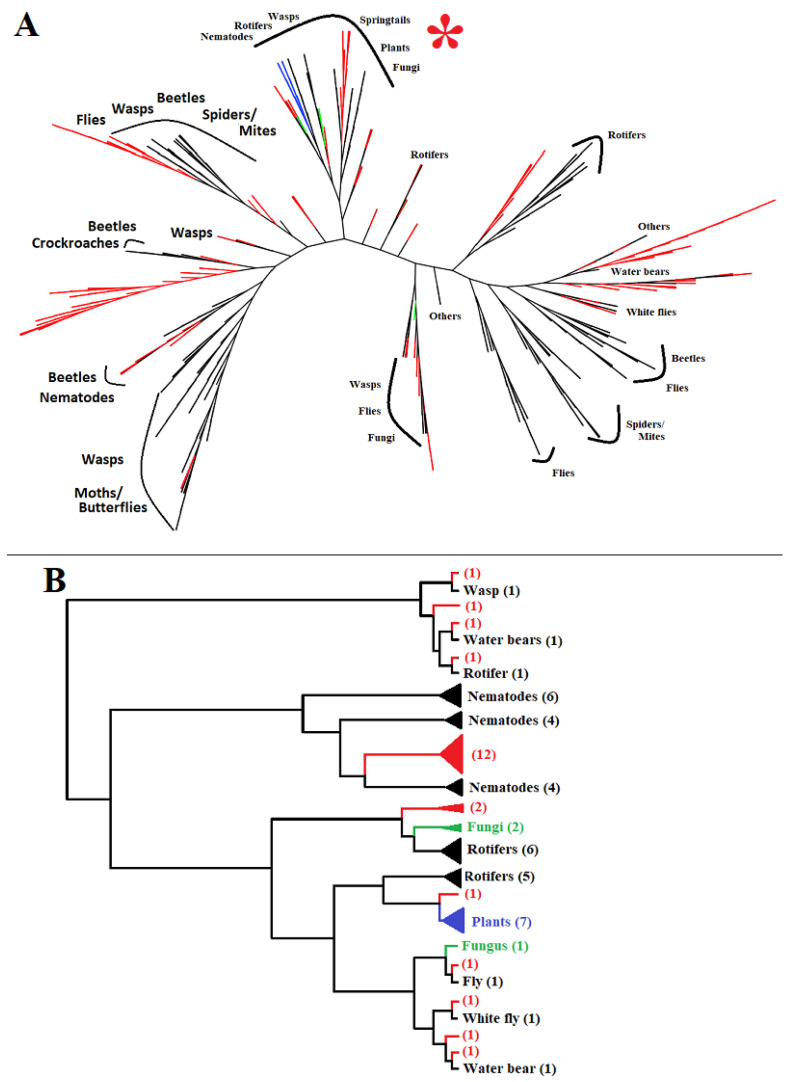
Phylogenetic relationship of the 242 GH32 proteins identified from animal genomes along with their best-matched bacterial proteins. (**A**): An overall phylogenetic tree. Red, blue, green, and black colors indicate bacterial, plant, fungal, and animal protein sequences, respectively. A single line could represent multiple sequences if they are closely related. The origin of the sequences is marked next to each branch. The branch marked by the symbol “*” is enlarged in panel B. (**B**): An enlarged view of one of the branches in panel A. Different colors represent different origins of the protein sequences. The sizes of the arrows are proportional to the number of the sequences in a sub-branch, which are indicated in the parentheses next to the origin of sequences.

**Figure 6 ijms-25-08296-f006:**
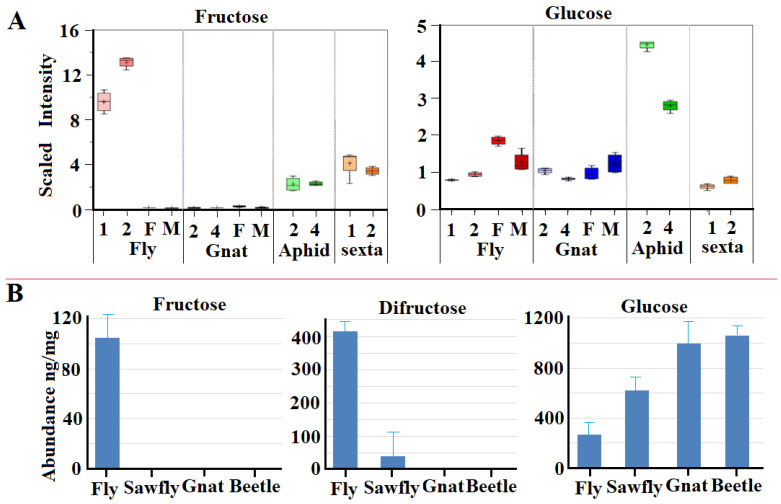
Abundance of fructose, difructose, and glucose among insect species with/without GH32 genes. (**A**): The relative abundance (expressed as scaled intensity) of fructose and glucose determined by a total metabolite profiling of whole insects. The numbers under the abscissa represent the instars of larvae and M and F represent males and females of adults. (**B**): Concentrations (ng/mg) of fructose, difructose, and glucose. The abbreviation of the insects are as follows: Fly, the Hessian fly, *Mayetiola destructor*; Gnat, the fungus gnat, *Bradysia* spp.; Aphid, the pea aphid, *Acyrthosiphon pisum*; Sexta, *Manduca sexta*; Sawfly, the wheat stem sawfly, *Cephus cinctus*; Beetle, the red flour beetle, *Tribolium castaneum*. The genomes of the Hessian fly and Manduca contain GH32 genes. The remaining insect species contain no GH32 genes.

**Table 1 ijms-25-08296-t001:** Putative GH32 genes identified from the genome sequences of different animals in Genbank *.

Animal Group	Scientific Name	Species with GH32	Genes Per Species	Total Genes
Dipterans	Diptera	7	1-12	35
Arachnids	Arachnida	11	1-5	21
Waters bears	Tardigrada	2	1, 6	6
Wasps	Hymenoptera	6	1-4	11
Springtails	Collembola	10	1-4	23
Rotifers	Bdelloida	8	1-7	34
Caddisflies	Trichoptera	4	1-5	9
Moths/Butterflies	Lepidoptera	11	1-9	44
Beetles	Coleoptera	9	1-4	24
Nematodes	Nematoda	5	1-7	16
Thrips	Thysanoptera	2	2-3	6
Millipedes	Diplopoda	1	1	1
Segmented worms	Annelida	1	1	1
Clam shrimps	Spinicaudata	1	1	1
Water fleas	Anomopoda	1	1	1
Woodlaces	Isopoda	1	1	1
Crabs	Pleocyemata	1	1	1
Cockroaches	Blattodea	1	3	3
True bugs	Hemiptera	1	2	2
Flatworms	Platyhelminthes	1	1	1
Sea spiders	Pycnogonida	1	1	1
Total		84		242

* species containing GH32 are as follows (numbers in parenthesis indicate the number of genes in this species). **Flies:** *Contarinia nasturtii* (12), *Mayetiola destructor* (10), *Bradysia coprophila* (2), *B. odoriphaga* (5), *Sitodiplosis mosellana* (4), *Phormia regina* (1), and *Bactrocera tryoni* (1). **Arachnids:** *Oedothorax gibbosus* (1), *Trichonephila clavipes* (1), *Tr. clavate* (1), *Caerostris darwini* (1), *Araneus ventricosus* (1), *Argiope bruennichi* (1), *Neoseiulus cucumeris* (3), *Oppiella nova* (2), *Leptotrombidium delicense* (5), *Tetranychus urticae* (2), and *Te. cinnabarinus* (2). **Water Bears:** *Ramazzottius varieornatus* (1) and *Hypsibius dujardini* (6). **Wasps:** *Tetragonula mellipes* (4), *Eupelmus annulatus* (3), *Macrocentrus cingulum* (1), *Trichogramma brassicae* (1), *Tr. evanescens* (1), and *Tr. pretiosum* (1). **Springtails:** *Sinella curviseta* (3), *Dicyrtomina minuta* (1), *Cyphoderus albinus* (3), *Pogonognathellus flavescens* (4), *Yoshiicerus persimilis* (2), *Tomocerus vulgaris* (2), *T. qinae* (2), *Allacma fusca* (2), *Orchesella cincta* (2), and *Folsomia candida* (2). **Rotifers:** *Macrotrachela quadricornifera* (7), *Rotaria* sp. Silwood1 (5), *Rotaria* sp. Silwood2 (6), *R. sordida* (1), *Didymodactylos carnosus* (7), *Adineta ricciae* (2), *A. vaga* (1), and *A. steineri* (5). **Caddisflies:** *Drusus annulatus* (3), *Micrasema longulum* (1), *Micropterna sequax* (1), and *Halesus radiatus* (5). **Moths and Butterlies:** *Helicoverpa armigera* (9), *Manduca sexta* (4), *Papilio polytes* (3), *Spodoptera exigua* (4), *Danaus plexippus* (3), *Amyelois transitella* (5), *Heliconius Melpomene* (1), *Bombyx mori* (2), *Pararge aegeria* (4), *Trichoplusia ni* (4), and *Ostrinia furnacalis* (5). **Beetles:**
*Listronotus oregonensis* (4), *Limonius californicus* (4), *Ignelater luminosus* (2), *Anoplophora glabripennis* (3), *Dendroctonus ponderosae* (3), *Rhynchophorus ferrugineus* (1), *Sphenophorus levis* (1), *Sitophilus oryzae* (2), and *Agrilus planipennis* (4). **Nematodes:** *Acrobeloides nanus* (2), *Heterodera schachtii* (2), *H. glycines* (1), *Meloidogyne enterolobii* (7), and *Globodera rostochiensis* (4). **Thrips:** Frankliniella occidentalis (3) and *Thrips palmi* (2). **True bugs:** *Bemisia tabaci* (1). **Cockroaches:**
*Blattella germanica* (3). **Millipede:** *Julidae* sp. (1). **Segmented worm:** *Streblospio benedicti* (1). **Clam shrimp:** *Eulimnadia texana* (1). **Water flea:** *Daphnia similis* (1). **Woodlace:** *Idotea baltica* (1). **Crab:** *Portunus trituberculatus* (1). **Flatworm:** *Schmidtea mediterranea* (1). Sea Spider: *Nymphon striatum* (1).

## Data Availability

All sequence data have been deposited here. Other data and information are provided here as Appendix A.

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
