# Peer review of "Frequent Acquisition of Glycoside Hydrolase Family 32 (GH32) Genes from Bacteria via Horizontal Gene Transfer Drives Adaptation of Invertebrates to Diverse Sources of Food and Living Habitats"

_ijms, 2024, doi:10.3390/ijms25158296_

Round 1

Reviewer 1 Report

Comments and Suggestions for Authors

The manuscript provide evidence of massive acquisition of the GH32 genes from bacteria via horizontal transfer. the results may help explaining some aspect of the biology  of certain animals and their interactions with host plants. 

I have few comments concerning the discussion.

The authors should speculate on the mechanisms that enabled transcription following horizontal gene transfer. Given that bacterial promoters significantly differ from eukaryotic promoters, it is crucial to address how transferred bacterial genes can be transcribed within a eukaryotic genomic environment. Some promoters, such as those from transposons (doi:10.3390/cells11030583) or viruses (doi:10.1038/srep25096), are known to drive transcription across distant phylogenetic systems.  It is possible that a new promoter was acquired with the aid of a transposon, or that it may have arisen de novo. Could the authors provide further discussion on this topic?

Additionally, what is the proposed vector responsible for the horizontal transfer of these sequences from bacteria to other organisms?

Some sentences are not clear.

l357 "...but the locations of the introns are totally different between the two genes in terms of protein coding" this statement is not clear. do you refer to intron position within the CDS?

Comments on the Quality of English Language

English is fine, few sentences need revision

Author Response

Please see point by point response to reviewers' comment in the attached resubmission letter

Reviewer 2 Report

Comments and Suggestions for Authors

Cheng et al. demonstrated that several invertebrate animals developed GH32 via multiple independent horizontal gene transfer from various bacteria. The authors retrieved GH32 sequences from animals by using TBLASTN program searching against WGS database, estimated their immediate ancestor by finding the best match bacterial sequences, and inferred their evolutionary history using phylogenetic analysis. They also conducted clustering analysis by CLANS program, indicating that GH32 could further be divided into two subclades. Additionally, they studied the intron number and position and compared the protein structure with bacterial version. Finally, they estimated the biological function of GH32.

Generally, Cheng et al. conducted a comprehensive analysis to study the HGT event of GH32. However, the order of the result presentation lacks a litter bit of logicality. Below is my suggestion. At first, the authors uncovered several GH32 homologs in animals (GH32 genes in diverse animals). After that, it is better to conduct structural analysis to show that these animal’s GH32 is real GH32. Then, using intron analysis to present the diversification of animal’s GH32. So that, it comes to a question: why they are divergent? Thus, the authors could try to detect their immediate bacterial ancestors and perform CLANS analysis to indicate that bacterial GH32 and animal’s GH32 belong to the same family.  Furthermore, the authors should infer their evolutionary history using phylogenetic analysis to show that there are multiple independent gene transfer events. Finally, the authors could estimate their biological function.

For the software or programs used in this study, please provide the references.

Some sentence are too wordy, please carefully polish the language.

Besides, I have some detailed suggestions.

Line37-89 (introduction): The authors provided detailed information about GH enzymes, the function of fructan, and HGT. However, paragraphs 1-3 appear to address three independent topics. Please refine the language and create a logical connection between them.

Line49-55: It's better to provide a few key examples.

Line59-65: It's better to provide a few key examples.

Line98: “H32”, do the authors mean “GH32”?

Line107-124: Why did the authors use tblastn against WGS database instead of using blastp against nr protein database? What is the advantage? Did the authors try to combine both the results of tblastn against WGS and the results of blastp against nr?

Line110: cite reference for tblastn program

Line126-127: which program did the author use to conduct the search? And cite reference for the program.

Line230: “we selected 11 representative lepidopterans to report in this study”, but in table1 row9, there are 10 species in the animal group of moths/butterflies (Lepidoptera). Why is there is different? Please double check these numbers.

Figure3: the legend shows that magenta dots represent either Nematoda or Rotifera. But in the figure, thers are several purple dots. Could the authors unify the color theme?

Figure4A: Could the authors give a more clear explanation for figure4A? What does the middle ruler represent? The numbers represent the number of genes? If one gene harbors more than two intron, how do you indicate it in this figure?

Line372-379: it is better to show the conservation pattern by presenting multiple sequences alignment.

Comments on the Quality of English Language

Some sentence are too wordy, please carefully polish the language.

Author Response

Please see item by item responses in the attached resubmission letter

Round 2

Reviewer 1 Report

Comments and Suggestions for Authors

The Authors have addressed the issues raised in my previous comments. 

However, I did not find a discussion regarding the vectors of horizontal transfer. I respectfully disagree with the authors' assumption that gene fragments can be directly incorporated into the genome of germline cells of the recipient species. This assumption implies that endosymbiont bacteria are involved as donor species. Could the authors provide further discussion on this topic?

Comments on the Quality of English Language

english language is fine

Author Response

Thanks.

Reviewer 2 Report

Comments and Suggestions for Authors

The authors have addressed all my concerns. I have no further comments.

Author Response

A couple of sentences have been added to the man manuscript to address the reviewer's comment. The sentences are:

"How GH32 genes were transferred from various bacteria to numerous animal species remains unknown.  The GH32 genes are in bacterial genomes without any obvious transposon elements around them.  Presumably, there were only two ways, either directly gene fragments transfer, or a gene integrated into a vector (plasmid or phages) first and then transferred into an animal germ cell via the vectors.  Further research needs to be done to delineate the transfer mechanisms." added to lines 544-549. 
